



# Contributions of equatorial planetary waves and small-scale convective gravity waves to the 2019/20 QBO disruption

Min-Jee Kang and Hye-Yeong Chun

Department of Atmospheric Sciences, Yonsei University, Seoul, South Korea

*Correspondence to*: Hye-Yeong Chun (chunhy@yonsei.ac.kr)

**Abstract.** In January 2020, unexpected easterly winds developed in the downward-propagating westerly quasi-biennial oscillation (QBO) phase. This event corresponds to the second QBO disruption in history, and it occurred four years after the first disruption that occurred in 2015/16. According to several previous studies, strong midlatitude Rossby waves propagating from the Southern Hemisphere (SH) during the SH winter likely initiated the disruption; nevertheless, the wave

forcing that finally led to the disruption has not been investigated. In this study, we examine the role of equatorial waves and small-scale convective gravity waves (CGWs) in the 2019/20 QBO disruption using MERRA-2 global reanalysis data. In June–September 2019, unusually strong Rossby wave forcing originating from the SH decelerated the westerly QBO at 0°–5°N at ~50 hPa. In October–November 2019, vertically (horizontally) propagating Rossby waves and mixed Rossby–gravity (MRG) waves began to increase (decrease). From December 2019, contribution of the MRG wave forcing to the zonal wind

deceleration was the largest, followed by the Rossby wave forcing originating from the Northern Hemisphere and the equatorial troposphere. In January 2020, CGWs provided 11% of the total negative wave forcing at ~43 hPa. Inertia–gravity (IG) waves exhibited a moderate contribution to the negative forcing throughout. Although the zonal-mean precipitation was not significantly larger than the climatology, convectively coupled equatorial wave activities were increased during the 2019/20 disruption. As in the 2015/16 QBO disruption, the increased barotropic instability at the QBO edges generated more

MRG waves at 70–90 hPa, and westerly anomalies in the upper troposphere allowed more westward IG waves and CGWs to propagate to the stratosphere. Combining the 2015/16 and 2019/20 disruption cases, Rossby waves and MRG waves can be considered the key factors inducing QBO disruption.

## 1 Introduction

The quasi-biennial oscillation (QBO) was first recorded through radiosonde wind observations in 1953 (Naujokat,

1986). Since then, a QBO phase transition has been made regularly by the descent of the opposite QBO phase with periods of 20–35 months. However, in February 2016, easterly forcing in the middle of the westerly winds disrupted the downward-propagating westerly QBO for the first time (Osprey et al., 2016), which is referred to as the 2015/16 QBO disruption. Because the QBO phase is highly correlated with extratropical/tropospheric phenomena, the impact of the disarrangement of



the westerly QBO phase by the sudden development of the easterly winds was not limited to the equatorial stratosphere
(Tweedy et al., 2017). The 2015/16 QBO disruption was primarily caused by equatorially propagating Rossby wave forcing.
The large magnitude of the Rossby wave flux in the Northern Hemisphere (NH) midlatitude (Osprey et al., 2016; Coy et al.,
2017; Hirota et al., 2018) and its increased amount of equatorward propagation by the strong subtropical westerlies in the
lower stratosphere (Barton and McCormack, 2017) likely induced the QBO disruption. However, the enhanced equatorial
wave forcing also contributed to the 2015/16 QBO disruption, which was first mentioned by Lin et al. (2019) and analyzed
in detail by Kang et al. (2020; KCG20 hereafter), who investigated each type of equatorial waves and small-scale convective
gravity waves (CGWs) during the 2015/16 QBO disruption.

        According to KCG20, inertia–gravity (IG) waves and mixed Rossby–gravity (MRG) waves at the altitude range of 40–
hPa in October–November 2015 preconditioned the zonal wind to be susceptible to the extratropical Rossby waves. In the
later stage, Rossby waves originating from the NH midlatitudes and the equatorial troposphere to the equatorial stratosphere
decelerated the QBO jet core, due to their considerably large magnitude compared to the climatology. In the final stage of
the disruption, the small-scale CGW forcing contributed to strengthening of the negative vertical wind shear by 20% of all
negative wave forcing. In October 2015–February 2016, stratospheric equatorial waves were unusually strong on account of
the exceptionally strong tropospheric convective activity. Moreover, the magnitude of westward-propagating IG waves and
CGWs was larger than that of the eastward waves probably due to the positive zonal wind anomalies at 70–200 hPa. The
strong MRG wave forcing was most likely generated from the increased barotropic instability at the QBO edges in the lower
stratosphere.

        Surprisingly, in January 2020, the westerly QBO phase was once again disrupted by the easterly winds at 43 hPa. This
occurrence suggests that the 2015/16 QBO disruption is not a single event and that QBO disruption may occur more
frequently in the future. Actually, the possibility of the second QBO disruption has already been raised by Raphaldini et al.
(2020), who demonstrated that the wind system related to an asymmetric zonal Rossby mode underwent a critical transition
(Dakos et al., 2012) around 2016. Anstey et al. (2020) suggested that large horizontal momentum flux in the Southern
Hemisphere (SH) propagating into the Tropics in June–September 2019 served as the most significant cause of the 2019/20
QBO disruption. The wave flux was not exceptionally strong after that period; however, the persistent wave forcing finally
disrupted the westerly winds at 43 hPa in January 2020. In the austral winter of 2019, Rossby wave activity in the
stratosphere was anomalously sufficiently strong to induce a minor sudden stratospheric warming (SSW) (Eswaraiah et al.,
2020; Shen et al., 2020). Therefore, it is likely that the strong extratropical Rossby waves during the SH winter initiated the
2019/20 QBO disruption. Nevertheless, a dominant wave forcing from October 2019 to January 2020, which finally reversed
the zonal wind sign, has not been examined yet, and the possible contributions from the equatorially trapped waves remain to
be investigated.

60        In this study, we provide a comprehensive overview of the 2019/20 QBO disruption by examining all the equatorial
waves (Kelvin, Rossby, MRG, and IG waves) and small-scale CGWs as in KCG20. To this end, we separate each equatorial



wave mode (Kim and Chun, 2015) and evaluate small-scale CGW forcing by using an offline CGW parameterization with Modern-Era Retrospective Analysis for Research and Applications version 2 (MERRA-2) reanalysis data (Gelaro et al, 2017). It should be noted that the same analysis tool and figure style as those in KCG20 are adopted to compare the 2015/16 and 2019/20 QBO disruptions. Section 2 describes the adopted reanalysis data and methods. Sect. 3 discusses the morphology of the equatorial waves and zonal wind (Sect. 3.1) and the quantitative estimation of each equatorial wave forcing and small-scale CGW forcing (Sect. 3.2) during the 2019/20 QBO disruption. In addition, the characteristics (including sources) of Rossby, MRG, IG waves, and small-scale CGWs are evaluated in Sect 3.3–3.6. Section 4 provides the concluding remarks.

## 2 Data and Methods

### 2.1 Reanalysis data

We use three-hourly output of MERRA-2 reanalysis data provided on a 0.5° latitude × 0.625° longitude grid at a native model-level from January 1980 to July 2020 (GMAO, 2015), using the same variables as in KCG20.

The 2019/20 QBO disruption was originally in the westerly QBO phase. In order to examine the difference between the climatological westerly QBO and the 2019/20 QBO disruption, we select the years with westerly QBO when the monthly mean zonal wind is greater than the monthly climatology by more than +0.5 standard deviation, both at 30 hPa and 50 hPa, for at least four months during the six months from April to September: 1980/81, 1985/86, 1990/91, 1993/94, 1995/96, 1997/98, 1999/2000, 2002/03, 2004/05, 2006/07, and 2011/12. This method ensures that the average of the 11 years, referred to in this study as the climatology, exhibits a downward QBO phase transition similar to that in 2019/20 (c.f., Fig. 3).

### 2.2 Methods

The temporal evolution of the zonal-mean zonal wind is investigated using the transformed Eulerian-mean (TEM) zonal momentum equation (Andrews et al., 1987):

$$\frac{\partial \bar{u}}{\partial t} = \left( f - \frac{1}{a \cos \phi} \frac{\partial}{\partial \phi} (\bar{u} \cos \phi) \right) \bar{v}^* - \bar{w}^* \frac{\partial \bar{u}}{\partial z} + \frac{1}{\rho_0 a \cos \phi} \nabla \cdot F + \bar{X} \,, \tag{1}$$

where $\bar{v}^*$ and $\bar{w}^*$ are defined by $\bar{v}^* = \bar{v} - \rho_0^{-1} (\rho_0 \overline{v'\theta'}/\bar{\theta}_z)_z$ and $\bar{w}^* = \bar{w} + (a \cos \phi)^{-1} \left( \cos \phi \, \overline{v'\theta'}/\bar{\theta}_z \right)_\phi$, which represent the residual meridional and vertical velocities, respectively. The term $\frac{1}{\rho_0 a \cos \phi} \nabla \cdot F$ represents the Eliassen–Palm flux (EPF) divergence (EPFD):





90
$$\frac{1}{\rho_0 a \cos\phi} \nabla \cdot F = \frac{1}{\rho_0 a \cos\phi} \left[ \frac{1}{a \cos\phi} \frac{\partial}{\partial\phi} \left( F^\phi \cos\phi \right) + \frac{\partial F^z}{\partial z} \right],$$
(2)

where $F^\phi$ [ $F^\phi = \rho_0 a \cos\phi \left( -\overline{u'v'} + \bar{u}_z \overline{v'\theta'}/\bar{\theta}_z \right)$ ] and $F^z$ [ $F^z = \rho_0 a \cos\phi (f - 1/(a \cos\phi)\, \partial/\partial\phi\, (\bar{u} \cos\phi)\, \overline{v'\theta'}/\bar{\theta}_z - $

$\overline{u'w'}$)] denote the meridional and vertical components of the EPF, respectively. The first and second terms of the $F^z$ are

referred to as $F^{z1}$ and $F^{z2}$, respectively. $\bar{X}$ term denotes the residual term, which includes the parameterized GWD.

95      In the equatorial region, the EPFD is calculated for each equatorial wave mode (Kelvin, Rossby, MRG, and IG waves).

The separation method is the same as that used in KCG20, following the method of Kim and Chun (2015). That is, in the

wavenumber–frequency ($k - \omega$) domain, spectral components with $|F^{z1}| < |F^{z2}|$ in the range of $0 < k \le 20$ and $\omega < 0.75$

cycle per day (cpd) in the symmetric spectrum are considered Kelvin waves, and the spectral components with $F^{z1} \times F^{z2} <$

0 in the range of $|k| \le 20$ and $0.1 \le \omega \le 0.5$ cpd in the antisymmetric spectrum are considered MRG waves. Among the

spectral components not classified as either of these wave types, those in the ranges of $|k| \le 20$ and $\omega \le 0.4$ are defined as

Rossby waves, with the remainder defined as IG waves. In the troposphere (below 100 hPa), IG waves are defined as (i) $|k|$

$> 20$ or (ii) $|k| \le 20$ and $\omega > 0.4$ cpd. The source level of the IG waves in the troposphere is assumed to be 140 hPa (c.f.,

KCG20). EPFD for each equatorial wave mode is calculated using Parseval's theorem.

To obtain small-scale CGW forcing constituting $\bar{X}$, an offline CGW parameterization is performed as in KCG20. First,

the phase-speed spectrum of the GW momentum flux generated from the diabatic forcing at the source level (cloud top) is

calculated. Second, the GW momentum flux and drag are calculated based on Lindzen's saturation scheme (Lindzen, 1981)

based on columnar propagation. It should be noted that, in order to constrain the magnitude of the CGW momentum flux

obtained from an offline parameterization to prevent over- or under-estimation of the CGW forcing, we use GWs observed

from super-pressure balloons in the tropical region (Jewtoukoff et al., 2013) (c.f., Kang et al., 2017). The small-scale CGWs

considered in this study have small horizontal wavelengths smaller than 100–200 km. The details of the parameterization

scheme of the CGWs can be found in KCG20.

As a key source of the equatorial waves, convective activity is investigated using the precipitation data provided by

MERRA-2. In addition, barotropic instability at the QBO edges is investigated as a potential source of the MRG waves

(Garcia and Richter, 2019; KCG20):


$$\bar{q}_\phi = 2\Omega \cos\phi - \left[ \frac{(\bar{u}\cos\phi)_\phi}{a \cos\phi} \right]_\phi - \frac{a}{\rho_0} \left( \frac{\rho_0 f^2}{N^2} \bar{u}_z \right)_z.$$
(3)

The negative regions of $\bar{q}_\phi$ indicate baroclinic/barotropic instability.





## 3. Results

### 3.1 Morphology of the zonal wind and each type of wave

Figure 1 shows the latitude–height cross section of the zonal-mean zonal wind from July 2019 to January 2020 with the corresponding monthly climatology (Fig. 1a) and the vertical profile of the zonal-mean zonal wind averaged for 5°N–5°S from July 2019 to January 2020 overlaid with the climatology (Fig. 1b). As early as July 2019, the northern side of the WQBO jet starts to be deformed. In September 2019, the westerly jet becomes weak at the altitude range of 40–50 hPa by

more than $1\sigma$ (Fig. 1b). Thereafter, the westerly wind at 43 hPa begins to decelerate, changing into the easterly in January 2020. The 2019/20 QBO disruption period shows a weaker westerly wind at altitudes near 30 hPa and a shallower WQBO jet compared to that in the 2015/16 QBO disruption period (Fig. 1 of KCG20). As in the 2015/16 QBO disruption, positive wind shear anomaly and westerly anomaly compared to the climatology are observed in the upper troposphere (100–150 hPa) in July–December 2019 and January 2020, respectively.

Figure 2 shows the EPF and EPFD for each equatorial wave and CGWs in a latitude–height cross section in January 2020. The EPF and EPFD are each multiplied by a factor of 2, except for the Rossby waves, to suitably represent the morphology of each wave. The P-CGWs (Fig. 2a) exhibit a positive (negative) forcing at 60–80 hPa (20–30 hPa and ~50 hPa), which is the strongest at 20 hPa over 5°N–5°S. Close to the equator, the negative CGW forcing is anomalously strong at 50–60 hPa.

In the lower stratosphere (60–100 hPa), Kelvin waves exert positive forcing on the QBO jet, thereby maintaining the westerly jet below the easterly wind development (Fig. 2b). However, the Kelvin wave forcing at 20–30 hPa is considerably smaller than that in February 2016 (Fig. 2b of KCG20); this is because the upper jet is very weak. The Kelvin waves propagating from the troposphere are larger than the climatology (Fig. S1), though the increase is lesser than that in January–February 2016.

MRG waves provide a strong negative forcing to the zonal wind at 25–100 hPa, concentrated at the equator (Fig. 2c). The negative MRG wave forcing at 40–50 hPa, which is critical for inducing the QBO disruption, is anomalously strong at 2°–5°N/S compared to the climatology. The MRG waves seem to be mainly generated at the location with positive EPFD in 5°–10°N/S and 60–90 hPa, as in the 2015/16 QBO disruption (Fig. 2c of KCG20).

IG wave forcing (Fig. 2d) shows negative values at 10°N–5°S, with an anomalously large magnitude located at 60–80

hPa and 5–15 hPa. In addition, Rossby wave forcing (Fig. 2e) exhibits large negative values at 0°–5°N, and they appear to propagate from the NH extratropics.

Figure 3 presents the monthly evolution of the zonal wind, zonal wind tendency, vertical advection (ADVz), required wave forcing (REQ), and each wave forcing averaged for 5°N–5°S from May 2019 to April 2020 at 10–70 hPa. In order to calculate the REQ, both the meridional and vertical advection terms are subtracted from the zonal-mean zonal wind tendency

in Eq. (1). From June to September 2019, the magnitude of the WQBO is reduced, without any significant downward





propagation (Fig. 3a), compared to the climatology (Fig. 3k). Comparison between the zonal-mean zonal winds in the 2019/20 QBO disruption and climatology (Fig. 3b) suggest an anomalous weakening of the zonal wind from July 2019, which is maximized at 40–60 hPa. The negative zonal wind tendency near 43 hPa is evident from June to August 2019 (Fig. 3c), which can be mainly attributed to the Rossby wave forcing (Fig. 3j).

The WQBO that maintains its depth without any significant downward propagation in June–September 2019 seems to be related to the strong ADVz (Fig. 3d). ADVz values at 20 hPa in June, July, August, and September 2019 are 9.6, 12.5, 13.3, and 11.3 m s month$^{-1}$, respectively, and these values are considerably larger than those for the climatology (2.8, 4.4, 5.8, and 6.3 m s$^{-1}$ mon$^{-1}$, respectively; Fig. 3m). In particular, the $\bar{w}^*$ values (Fig. S2) in July and September 2019 are 0.7 and 0.9 mm s$^{-1}$, respectively, which are 1.6 and 1.5 times larger than that for the climatology, respectively. In this period, midlatitude
Rossby wave forcing is extremely large and induces a minor SSW (Anstey et al., 2020; Eswaraiah et al., 2020; Shen et al., 2020), possibly resulting in the enhanced vertical upwelling of the Brewer–Dobson circulation (BDC) and, thereby, a large magnitude of the ADVz. This implies that the ADVz can help QBO disruption by retarding the downward propagation of the WQBO jet. Although the 2019 SSW is classified as a minor SSW in that the zonal-mean zonal wind at 10 hPa at 60°S does not undergo a reversal, the zonal wind at ~32 km at 72°S shows an easterly wind (Eswaraiah et al., 2020), which implies a
strong Rossby wave forcing in the SH.

Climatologically, REQ (Fig. 3n) exhibits a negative (positive) sign in negative (positive) wind shear zone, and the sign of P-CGW forcing (Fig. 3f) generally follows that of the REQ. The larger the magnitude of the vertical wind shear, the more the P-CGWs explain the REQ. However, in June–July–August (JJA) 2019 (Fig. 3e), a negative REQ is observed at 30–60 hPa without negative vertical wind shear; this seems to be unusual. The P-CGWs start to contribute to the deceleration of the
QBO jet after the negative vertical wind shear is generated at ~50 hPa (i.e., October 2019). In contrast to the strong Kelvin wave forcing in the 2015/16 QBO disruption, Kelvin wave forcing (Fig. 3g) in the 2019/20 QBO disruption is smaller than or comparable to the climatology (Fig. 3p). This weak Kelvin wave forcing could be one of the reasons why the upper jet at 20–30 hPa is not maintained after the QBO disruption.

During the 2019/20 QBO disruption, the momentum forcing by the MRG waves (Fig. 3h) is considerably stronger than
its climatology (Fig. 3q). For instance, from October 2019 to January 2020 the MRG wave forcing at 43 hPa is dominant among that of the equatorial waves, largely explaining the REQ. This result suggests that MRG waves play a role in reversing the sign of the zonal in the later stages. IG wave forcing (Fig. 3i) shows strong negative values in May 2019 above 43 hPa and after July 2019 following the negative wind shear zone. Rossby wave forcing (Fig. 3j) is strong from June to September 2019 below ~20 hPa. At 40–50 hPa, Rossby waves continue to provide a negative wave forcing until February
180 2020.





### 3.2 Contributions of each wave type at 43 hPa

Figure 4 shows the monthly evolution of zonal wind, zonal wind tendency, and wave forcing of each wave type from May 2019 to April 2020 at 43 hPa; their exact values and percentages are summarized in Table 1. As early as May 2019, the zonal wind tendency (dotted line in Fig. 4a) becomes negative, while, in January 2020, the zonal wind (solid line in Fig. 4a)

becomes easterly. The negative wind tendency is weakened until October 2019 although it intensifies again in November 2019. The negative wind tendency in May 2019 is mainly explained by the Rossby (-0.62 m s$^{-1}$ mon$^{-1}$) and IG (-0.57 m s$^{-1}$ mon$^{-1}$) waves, with contributions of 48% and 45%, respectively. The momentum forcing by the Rossby waves becomes dominant from June to November 2019. The maximum contribution is 82% (in July 2019), and it decreases subsequently. In December 2019 and January 2020, the MRG wave forcing accounts for 44% and 41% of the total negative wave forcing,

respectively, which are larger than any other equatorial wave forcing. During the same period, the Rossby wave forcing is the second largest, with contributions of 33% and 38%, respectively. In January 2020, parameterized CGWs start to contribute to the easterly development (11%), and they provide large negative forcing in February 2020 with a percentage of 44%.

The contribution from the parameterized CGWs is smaller than that in the 2015/16 QBO disruption. As shown in Fig.

13, the magnitude of the source-level westward CGW momentum flux is not significantly larger than that of the climatology; this is the probable cause of the smaller magnitude of the negative CGW forcing during the 2019/20 disruption than that during the 2015–2016 disruption. The smaller CGW forcing is also explained by the vertical wind shear at ~40 hPa in January 2019 (Fig. 4d) being smaller than in February 2016 (Fig. 4d of KCG20).

The meridional and vertical EPFD of the Rossby waves at 43 hPa are shown in Fig. 4c. In May–September 2019, the

meridional component dominates the total Rossby wave forcing, which confirms the strong meridional propagation of the Rossby waves from the SH midlatitudes during the austral winter (Anstey et al., 2020). However, in November 2019–February 2020 (i.e., boreal winter) the meridional component becomes stronger, and its magnitude is comparable to that of the vertical component.

In summary, the negative forcing by the Rossby waves contributes most to the zonal wind deceleration from June to

September 2019. MRG wave forcing intensifies from October 2019, and it becomes the strongest among all the equatorial wave forcings in December 2019–January 2020. IG waves decelerate the WQBO jet with a moderate magnitude throughout, and the P-CGWs contribute 11% of the negative forcing in January 2020.

### 3.3 Rossby waves

Figure 5 shows the latitude–height cross sections of the EPF and EPFD for the Rossby waves and the corresponding

meridional and vertical components in July 2019, August 2019, October 2019, and January 2020. The meridional EPF (EPF-y) values at 10°N and 10°S are presented on the left and right sides of the EPFD-y, respectively, and the vertical EPF (EPF-z) at 70 hPa is presented at the bottom of EPFD-z using red lines. The climatology is represented by black lines, with the ±1σ





indicated by the gray shading. In July 2019 (Fig. 5a), the EPFD for the Rossby waves is unusually strong at the northern flank of the QBO at 40–60 hPa. The meridional EPFD dominates the total EPFD at 40–60 hPa in the NH. They are most

likely to propagate from the SH based on the large northward EPF at 10°S. Moreover, vertical EPF at 70 hPa is larger than the climatology at 10°N–10°S; accordingly, a large negative EPFD-z can be observed at 30–50 hPa. In August 2019 (Fig. 5b), there is evident deceleration of the WQBO jet by the Rossby waves propagating from the SH; however, the negative wave forcing becomes stronger at the jet core. It is also found that the EPF-z at 70 hPa in August 2019 is larger than that in July 2019 at 5°N–20°S.

In October 2019 (Fig. 5c), the shape of the zonal wind is significantly deformed by the anomalously strong negative forcing in the WQBO jet, mainly attributed to the strong meridional Rossby wave forcing originating from the SH. In January 2020 (Fig. 5d), when the QBO disruption occurs, the Rossby wave forcing is generally weaker than that shown in Figs. 5a–c; consequently, the EPFD in Fig. 5d is multiplied by a factor of two. The Rossby waves laterally propagating from the NH decelerate an isolated small westerly jet at 30–40 hPa, while the vertically propagating Rossby waves provide an

anomalously strong easterly forcing below the altitude of 40 hPa at 25°N–15°S, except close to the equator. The EPF-z at 70hPa, which is larger during the disruption period than the climatology at 0°–20°S and 10°–20°N, confirms the presence of the strong Rossby waves propagating from the equatorial region.

In summary, Rossby wave forcing and flux during the austral winter of 2019 have a dominant meridional component propagating from the SH. However, a relatively small magnitude of the Rossby wave forcing is found with comparable

meridional and vertical components in January 2020. The strong EPF-z at 70 hPa mostly propagates from the equatorial troposphere and the NH, when the EPF is traced back to the troposphere (Fig. S3).

As mentioned previously, a minor SSW took place in the SH in September 2019, which was an exceptionally rare event. This implies that Rossby wave flux and forcing in the midlatitude stratosphere was above average during the austral winter of 2019. Figure 6 shows the latitude–height cross section of the EPF overlaid with the zonal-mean zonal wind (Fig. 6a),

vertical EPF at 100 hPa (Fig. 6b), and zonal wind at 15°S (Fig. 6c) in JJA. The red line represents the 2019 case, and the black line represents the climatology. The waves are generally vertically propagating, while a part of the waves propagates into the Tropics. The vertical EPF penetrating the stratosphere is considerably larger than the climatology by ~2σ (Fig. 6b). An excessively large EPF in the midlatitude stratosphere could also propagate into the equator because the zonal-mean zonal wind in the SH subtropics at 40–80 hPa exhibits stronger westerly winds than the climatology (Fig. 6c).

**3.4 MRG waves**

Figure 7 shows the EPF and EPFD similar to Fig. 5 but for the MRG waves in October, November, and December 2019 and January 2020. In October 2019 (Fig. 7a), the MRG waves exert strong negative forcing, especially at 20–50 hPa between 5°N and 5°S, and at 10–40 hPa between 5°N and 10°N. The negative MRG wave forcing at 30–50 hPa near the equator, which is strongly related to the QBO disruption, seems to propagate from the regions with positive EPFD: (i) 60–80 hPa at





5°–10°N, (ii) 40–80 hPa near 10°N, and (iii) ~40 hPa near 10°S. This is supported by considerably stronger vertical EPF at 70 hPa at 0°–10°N and meridional EPF at 20–50 hPa at 10°N/S. In November 2019 (Fig. 7b), similar features as in October 2019 are shown but with a reduced vertical range for the negative wave forcing near the equator.

In December 2019 (Fig. 7c), westerly winds at 30–50 hPa are weakened. The negative MRG wave forcing becomes unusually strong at 50 hPa in the 5°–10°S range, although the increase in the EPF-z at 70 hPa is smaller than those in

October and November 2019. In January 2020 (Fig. 7d), MRG wave forcing at 43 hPa is the largest among all the equatorial wave forcings. Not only the equatorward waves at 10°N/S at 30–50 hPa but also the equatorward and upward waves at 10°N/S at 70 hPa are much stronger than the climatology by more than $1\sigma$. In particular, the upward and equatorward EPF vectors starting from 5°–10°S at 70 hPa appear to exhibit the maximum contribution to the negative forcing observed at 43 hPa.

Figure 7 shows that the MRG waves weaken the QBO jet and finally reverse the wind sign in the later period (e.g., December 2019 and January 2020). The negative MRG wave forcing is exerted on the jet core not only at the 43 hPa altitude but also at the altitude range of 25–50 hPa, resulting in an excessive weakening of the upper jet (~30 hPa) during the 2019/20 QBO disruption. In addition, MRG waves are strongly generated in regions with a large horizontal wind curvature, coincident with the location of the positive EPFD. Therefore, in order to investigate whether the MRG waves are generated

by barotropic/baroclinic instability, we select a region (boxed region in Fig. 8) with small positive $\bar{q}_\phi$ values.

Figure 8 shows the monthly-averaged $\bar{q}_\phi$ and the daily time series of the number of grids with the negative $\bar{q}_\phi$ at the boxed region in December 2019 (Fig. 8a,c) and January 2020 (Fig. 8b,d), along with the climatology. Note that the total number of grids in the boxed region is 33. The monthly mean $\bar{q}_\phi$ in the boxed region shows small positive values in December 2019 and in January 2020; however, the number of negative $\bar{q}_\phi$ in the boxed region based on the daily-mean

values (Figs. 8c–d) is generally much larger during the disruption period compared to that of the climatology. The barotropic term [first two terms on the right-side of Eq. (3)] dominates the $\bar{q}_\phi$ value in the boxed region; on that basis barotropic instability at the boxed region is likely to generate anomalously strong MRG waves.

Figure 9 represents the zonal-mean precipitation in the tropical region. Generally, the precipitation from June 2019 to January 2020 is comparable to the climatology, except for that in June and October 2019 at 5°N–5°S. In June 2019 and

October 2019, the precipitation is greater than the climatology by ~$1\sigma$, corresponding to a much weaker enhancement compared to that in the 2015/16 QBO disruption.

Now we examine the precipitation spectrum in association with the equatorial wave mode during the 2019/20 disruption. Figure 10 shows 10°S to 10°N averaged precipitation spectrum as a function of zonal wavenumber ($k$) and frequency ($\omega$), divided by the background spectrum for the symmetric (left) and antisymmetric (right) components with respect to the

equator from October 2019 to January 2020. The background spectrum of the symmetric (antisymmetric) component is obtained by applying 1-2-1 smoothing for $k$ and $\omega$ 40 and 10 times, respectively, to the raw symmetric (antisymmetric) spectrum (c.f. KCG20). Following Wheeler and Kiladis (1999), the values greater than 1.4 in Fig. 10 are considered as





statistically significant wave signals at the 95% confidence level. The spectrum more than $1\sigma$ stronger than the climatology (blue-stippled pattern) starts to widen in December 2019, although the area is smaller than that in the 2015/16 QBO disruption. Generally, the strong power is evident in the spectrum related to the Kelvin and IG waves. In the symmetric spectrum, statistically significant Rossby wave signals ($k$ = -16–19, $\omega$ = 0.06–0.1 cpd) are shown, which are stronger than the climatology by more than $1\sigma$ in November 2019–January 2020 (Figs. 10b–d). The enhancement of the Rossby waves in the troposphere in January 2020 probably affects the large vertical EPF at 70 hPa (Fig. 5d). Kelvin wave signals ($k$ = 0–8 and $\omega$ = 0–0.25) are statistically significant throughout and are more than $1\sigma$ stronger than the climatology after November 2019. It is likely that these waves propagate to the stratosphere and, thereby, contribute to the strong EPF-z at 70 hPa (see Fig. S1). In the antisymmetric spectrum, the MRG wave signals in the antisymmetric spectrum ($k$ = -10–0 and $\omega$ = 0.2–0.32) are stronger than the climatology by more than $1\sigma$ in December 2019–January 2020. Therefore, the enhanced convective activity in the MRG wave spectrum in December 2019–January 2020, together with the barotropic instability at the QBO edges, may affect the anomalously strong MRG wave forcing near 43 hPa. Overall, convectively coupled equatorial waves are slightly enhanced in the later period of the 2019/20 QBO disruption, although the zonal-mean precipitation is not significantly increased.

### 3.5 IG waves

Figure 11 shows the EPF and EPFD as a function of latitude and height, and latitudinal distribution of the vertical EPF by the IG waves at 70 hPa from October 2019 to January 2020. Given that the IG waves generally propagate upward in the stratosphere, the upward-directed EPF vectors inside the WQBO jet at 5°N–5°S indicate a larger magnitude of the westward IG waves compared to that of the eastward IG waves. The negative IG wave forcing is exerted on the jet core throughout, with a significant magnitude located at the altitude range of 60–90 hPa. However, the magnitude of the EPF-z at 70 hPa is slightly larger than that of the climatology in December 2019 and January 2020, differing from the case in the 2015/16 QBO disruption.

Figure 12 illustrates the 10°S to 10°N averaged phase-speed spectrum of the precipitation for the IG wave ranges, which approximately represents the source spectrum of the IG waves in December 2019 (Fig. 12a) and January 2020 (Fig. 12b) along with the climatology. Generally, the disruption period shows a larger IG wave source spectrum by ~$1\sigma$ compared to the climatology. The zonal wind speed at 140 hPa is approximately 2.6 m s$^{-1}$ and 4.9 m s$^{-1}$ in December 2019 and the climatology, respectively. Therefore, the IG source spectra during both the disruption period and climatology exhibit dominant westward components, although the climatology exhibits additional westward waves in the phase speed of 2.6–4.9 m s$^{-1}$. However, the additional westward waves of the climatology in 2.6–4.9 m s$^{-1}$ are dissipated by the critical-level filtering (-0.2–5.4 m s$^{-1}$), and this range is wider than that (1.1–4.2 m s$^{-1}$) of the disruption period. Thus, the remaining westward waves at 70 hPa are stronger in December 2019 than the climatology. The narrower critical-level filtering range is related to the westerly anomalies and easterly anomalies at 70–100 hPa and 100–140 hPa, respectively (Fig. S4). In January



2020, compared to the climatology, the estimated IG spectrum at the source level exhibits additional westward waves in 2.2–4.4 m s$^{-1}$ due to the stronger westerlies at the source level (Fig. S4). Despite these waves being almost filtered by the critical-level filtering process, the eastward shift of the critical-level filtering range compared to the climatology results in more westward waves remaining at the altitude of 70 hPa. In addition, the eastward waves in the climatology are less filtered than those during the disruption period. The findings shown in Fig. 12 indicate that slightly strong westward IG waves at 70 hPa

during the disruption period can be explained by the narrow critical-level filtering range for the westward IG waves and the enhanced convective activity. This conclusion is similar to that regarding the IG waves during the 2015/16 QBO disruption.

### 3.6 Parameterized CGWs

Figure 13 presents the 5°N–5°S averaged zonal-mean zonal wind and CGWD (top) and the source-level CGW momentum flux (i.e., cloud-top momentum flux; CTMF) (bottom) in January 2020, along with the climatology. The

maximum negative CGWD of -0.7 m s$^{-1}$ mon$^{-1}$ is shown at 47 hPa, where there is negative vertical wind shear; This magnitude is less than the half of the maximum negative CGWD in February 2016. The westward-propagating CTMF is comparable to the climatology, consistent with the small negative CGWD.

Figure 14 shows the convective source spectrum and the wave-filtering and resonance factor (WFRF) spectrum in January 2020 and the climatology. As mentioned in KCG20, the CTMF spectrum is derived based on the spectral

combination of the convective source spectrum and the WFRF [see Eq. (1) of Kang et al., 2017]. The convective source spectrum is amplified at the phase velocity equal to the convection moving speed ($c_{qh}$), and its overall magnitude is dependent on the square of the convective heating rate. The following effects are included in the WFRF: (i) critical-level filtering within the convection and (ii) resonance between the vertical harmonics constituting convective forcing and the natural wave modes given by the dispersion relationship (Song and Chun, 2005; KCG20). The convective source spectrum

(Fig. 14a) is slightly stronger than that for the climatology owing to the slightly stronger convection during the disruption. The WFRF (Fig. 14b) is also slightly stronger with a slightly wider spectrum than that for the climatology in 5°N–5°S. Therefore, both the convective source spectrum and WFRF lead to a somewhat stronger CTMF compared to the climatology. Furthermore, as 2019 was recorded as the second-warmest year (GISTEMP 2020), global warming likely led to higher static stability at the cloud top and, hence, to the strong CTMF. This is because the CTMF generally increases as the stability

increases due to the proportionality of the stability at and above the cloud top to the CTMF [$N_2$ is proportional to $\bar{M}_c$ in Eq. (22) in Song and Chun (2005)]. However, the enhancement of the CTMF by CGWs in 2019/20 is much smaller than that in the 2015/16 QBO disruption.

In Fig. 14, white and gray line represents the zonal wind at the cloud top ($U_{ct}$) and the moving speed of convection ($c_{qh}$), respectively. The $U_{ct}$ averaged for 5°N–5°S exhibits a weaker easterly (-3.0 m s$^{-1}$) compared to the climatology (-4.5

m s$^{-1}$). In addition, $c_{qh}$ exhibits a weaker easterly (-2.1 m s$^{-1}$) compared to the climatology (-2.8 m s$^{-1}$). The eastward shifts of the zonal wind at the cloud top and $c_{qh}$ cause stronger westward and eastward momentum fluxes, respectively; the





competition between the two factors results in an increased eastward momentum flux (bottom panel of Fig. 13). In our CGW parameterization, we obtain $c_{qh}$ by averaging the zonal wind below 700 hPa, which is related to the propagation speed of the gust front (Choi and Chun, 2011). Therefore, the westerly anomalies in the $c_{qh}$ are caused by the westerly anomalies in the

zonal wind below 700 hPa. The westerly anomalies in the lower troposphere frequently occur under El Niño conditions and in future climate simulations (Lu et al., 2008; Collins et al., 2010; Kawatani et al., 2019). Therefore, the high surface temperature during the 2019/20 QBO disruption likely led to the westerly anomalies in the lower troposphere. There is a need for further study on the cause and significance of the westerly anomalies in a warmer climate, although doing so is beyond the scope of this study. Although the magnitude of the westward CGWs at the source level is similar to that in the

climatology, the eastward shift of the zonal winds at 100–200 hPa (Fig. 13) resulted in more westward waves propagating into the stratosphere compared to those in the climatology. Overall, the increase in the CGW momentum flux in January 2020 is considerably smaller than that in February 2016, and no significant increase is observed in the westward momentum flux. Together with the weaker negative vertical wind shear at 43 hPa, this results in a small magnitude of the negative CGW forcing near 43 hPa.

**4. Summary and Conclusions**

In this study, we examined the role of each equatorial planetary wave mode and parameterized convective gravity waves (CGWs) in the 2019/20 QBO disruption and compared with the results from the 2015/16 QBO disruption (KCG20). Using MERRA-2 model-level data, we separated each equatorial wave mode (Kim and Chun, 2015) and obtained small-scale CGW forcing by performing an offline CGW parameterization (Kang et al., 2017). The main results are summarized

schematically in Fig. 15 and in the following text:

- From June to September 2019, unusually strong Rossby wave forcing at ~50 hPa decelerated the westerly QBO jet at 0°–5°N. The strong Rossby wave flux propagated mostly from the SH midlatitudes due to the large wave activity associated with the 2019 minor SSW in the SH and the westerly anomalies in the SH subtropics. MRG and IG wave forcing partly contributed to the wind deceleration.

- From October to November 2019, laterally propagating Rossby wave flux from the SH was weakened, with the vertically propagating Rossby wave flux from the Tropics being enhanced. MRG wave forcing increased with nearly the same contribution as that from the latitudinally propagating Rossby waves. Furthermore, the IG wave forcing began to increase, albeit with a smaller magnitude than that of the MRG wave forcing. In this period, the oval structure of the zonal wind was significantly deformed.

- From December 2019 to January 2020, the momentum forcing by the MRG waves was stronger than that by any other equatorial waves, mainly due to the strong barotropic instability at the QBO edges at 70–90 hPa, and partly due to the enhanced convective activity, as in the 2015/16 QBO disruption. Rossby waves propagating from the NH midlatitudes also decelerated the QBO jet. In January, the QBO westerly was changed to easterly at 43 hPa.





The CGWs strengthen the negative wind shear near the equator by exerting negative forcing at 40–50 hPa by 11%
of the total negative wave forcing. The negative CGWD in this period did not show a significant increase due to a
less evident increase in the convective activity and eastward shift of the convection moving speed compared to the
climatology. In this period, the magnitude of the westward IG wave momentum flux was slightly larger than that
of the climatology at 70 hPa, owing to slightly stronger convection and the narrower critical-level filtering range.

- From November 2019 to January 2020, the Kelvin waves, and partly the CGWs, exert positive forcing on the
westerly QBO wind at 60–80 hPa. This finding is important, as it implies that the zonal wind at 60–80 hPa could
have been decelerated by the negative wave forcing in the absence of the positive momentum forcing.

Compared to the 2015/16 QBO disruption, the 2019/20 QBO disruption exhibited weaker and thinner westerly winds
near 30 hPa. Therefore, at first glance, the 2019/20 QBO disruption appears as a normal QBO, propagating downward with
time. This is because Rossby waves propagating from the midlatitudes, which induce a localized wind deceleration, were the
strongest in the early stage of the 2019/20 QBO disruption. In the later stage, vertically MRG wave forcing mainly induced
the wind reversal unlike in the 2015/16 QBO disruption. Since the MRG wave forcing did not provide a localized wave
forcing, the large magnitude of the MRG wave forcing resulted in a deceleration of the entire westerly jet above the altitude
of 43 hPa. Alternative reasons for inducing weaker and thinner westerly winds at 20–30 hPa include relatively shallower
QBO depth and weaker positive wave forcing (i.e., Kelvin waves and eastward CGWs).

It is interesting that the midlatitude Rossby waves intruded into the Tropics when the tropical vertical upwelling was
exceptionally strong (February 2016; August–September 2019). This relationship appears to be intuitive, because a strong
midlatitude Rossby wave forcing in the stratosphere drives a strong BDC. The instantaneous upward extension of the
WQBO due to the strong BDC likely facilitated the QBO disruption by preventing the negative wave forcing from
decelerating the top/bottom of the QBO. Therefore, the tropical branch of the BDC and its possible influence on the 2015/16
and 2019/20 QBO disruptions should be further examined.

The 2019/20 QBO disruption occurred under the following conditions: (i) strong horizontal component of the Rossby
wave forcing that originated from the SH in the early stages, (ii) strong MRG wave forcing generated from the barotropic
instability at the QBO edges in the later stages, and (iii) negative IG and CGW forcing due to the slightly enhanced
convective activity and westerly anomalies in the UTLS. Therefore, the westerly anomalies in the subtropics/tropics and the
strong baroclinic instability in the lower stratosphere mainly led to anomalously strong wave forcing, which in turn led to the
QBO disruption. The findings of this study and KCG20 indicate considerable differences in the temporal evolutions of the
wave forcing driving the 2015/16 and 2019/20 QBO disruptions. However, both disruptions involved significant
contributions from the midlatitude Rossby waves under the environmental conditions that are favorable for equatorward
propagation and the MRG waves that are generated in situ from the barotropic instability. In this regard, a better
understanding of the two wave modes can help enhance the predictability of the QBO disruption and the associated
atmospheric phenomena in the troposphere (e.g., Madden–Julian oscillation). More frequent occurrence of the QBO



disruptions in the future has been suggested by previous studies mainly due to the increase in the Rossby wave flux propagating toward the equator and weakening of the QBO amplitude with the climate changes. Moreover, considering the large contribution of the equatorial planetary and gravity waves in the two QBO disruption cases, it is also necessary to
investigate how these waves will change with climate change and how this change will affect the more frequent occurrence of QBO disruptions.

*Data availability.* The MERRA-2 data are available at the Global Modeling and Assimilation Office at NASA Goddard Space Flight Center through the NASA GES DISC online archive (available online at https://gmao.gsfc.nasa.gov/reanalysis/,
GMAO, 2015).

*Author contributions.* HYC and MJK conceived the study, and MJK perform it. MJK drafted the manuscript with a contribution from HYC.

*Competing interests.* The authors declare they have no conflict of interest.

*Acknowledgments.* This work was supported by the National Research Foundation of Korea (NRF) grant funded by the Korea government (MSIT) (No. 2020R1A4A1016537).

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



**Table 1.** Monthly-averaged momentum forcing by each wave type (m s$^{-1}$ month$^{-1}$) at 43 hPa averaged for 5°N–5°S from June to January for the disruption period (2019/20) and the climatology. The ratio of each wave forcing to the total negative forcing is given in the parenthesis only when the wave forcing is negative.

| 2019/20 | Jun 2019 | Jul 2019 | Aug 2019 | Sep 2019 | Oct 2019 | Nov 2019 | Dec 2019 | Jan 2020 |
|---|---|---|---|---|---|---|---|---|
| MRG | -0.4 (12%) | -0.3 (8%) | -0.4 (14%) | -0.7 (27%) | -0.6 (30%) | -0.7 (34%) | -1.1 (44%) | -1.2 (41%) |
| IG | -0.5 (15%) | -0.3 (10%) | -0.5 (15%) | -0.4 (14%) | -0.4 (21%) | -0.5 (24%) | -0.5 (23%) | -0.3 (10%) |
| Rossby | -2.2 (73%) | -2.8 (82%) | -2.4 (71%) | -1.6 (59%) | -0.9 (49%) | -0.8 (42%) | -0.8 (33%) | -1.1 (38%) |
| CGW | 0.6 | 0.6 | 0.7 | 0.7 | 0.9 | 0.5 | -0.01 (0%) | -0.3 (11%) |
| Kelvin | 1.3 | 1.5 | 1.8 | 1.7 | 1.3 | 1.1 | 0.9 | 0.4 |
| Rossby-Y | -1.8 (58%) | -2.2 (65%) | -1.7 (51%) | -1.1 (41%) | -0.6 (32%) | -0.4 (22%) | -0.4 (17%) | -0.5 (19%) |
| Rossby-Z | -0.5 (15%) | -0.6 (17%) | -0.7 (20%) | -0.5 (18%) | -0.3 (17%) | -0.4 (20%) | -0.4 (16%) | -0.5 (19%) |
| **Climatology** | **Jun** | **Jul** | **Aug** | **Sep** | **Oct** | **Nov** | **Dec** | **Jan** |
| MRG | -0.2 (16%) | -0.1 (5%) | -0.1 (5%) | -0.1 (10%) | -0.2 (19%) | -0.4 (23%) | -0.5 (20%) | -0.4 (19%) |
| IG | -0.1 (9%) | -0.3 (15%) | -0.3 (18%) | -0.4 (33%) | -0.7 (54%) | -0.9 (48%) | -0.9 (39%) | -0.8 (36%) |
| Rossby | -1.2 (75%) | -1.7 (80%) | -1.4 (77%) | -0.7 (57%) | -0.3 (27%) | -0.4 (24%) | -0.6 (26%) | -0.7 (34%) |
| CGW | 1.0 | 0.8 | 0.6 | 0.6 | 0.4 | -0.1 (5%) | -0.3 (14%) | -0.2 (11%) |
| Kelvin | 2.2 | 1.3 | 1.0 | 0.9 | 0.7 | 0.5 | 0.6 | 0.7 |
| Rossby-Y | -1.1 (72%) | -1.6 (74%) | -1.2 (70%) | -0.6 (51%) | -0.1 (12%) | 0.0 (0%) | -0.0 | -0.1 (5%) |
| Rossby-Z | -0.1 (3%) | -0.1 (6%) | -0.2 (7%) | -0.1 (6%) | -0.2 (15%) | -0.4 (24%) | -0.6 (26%) | -0.6 (29%) |




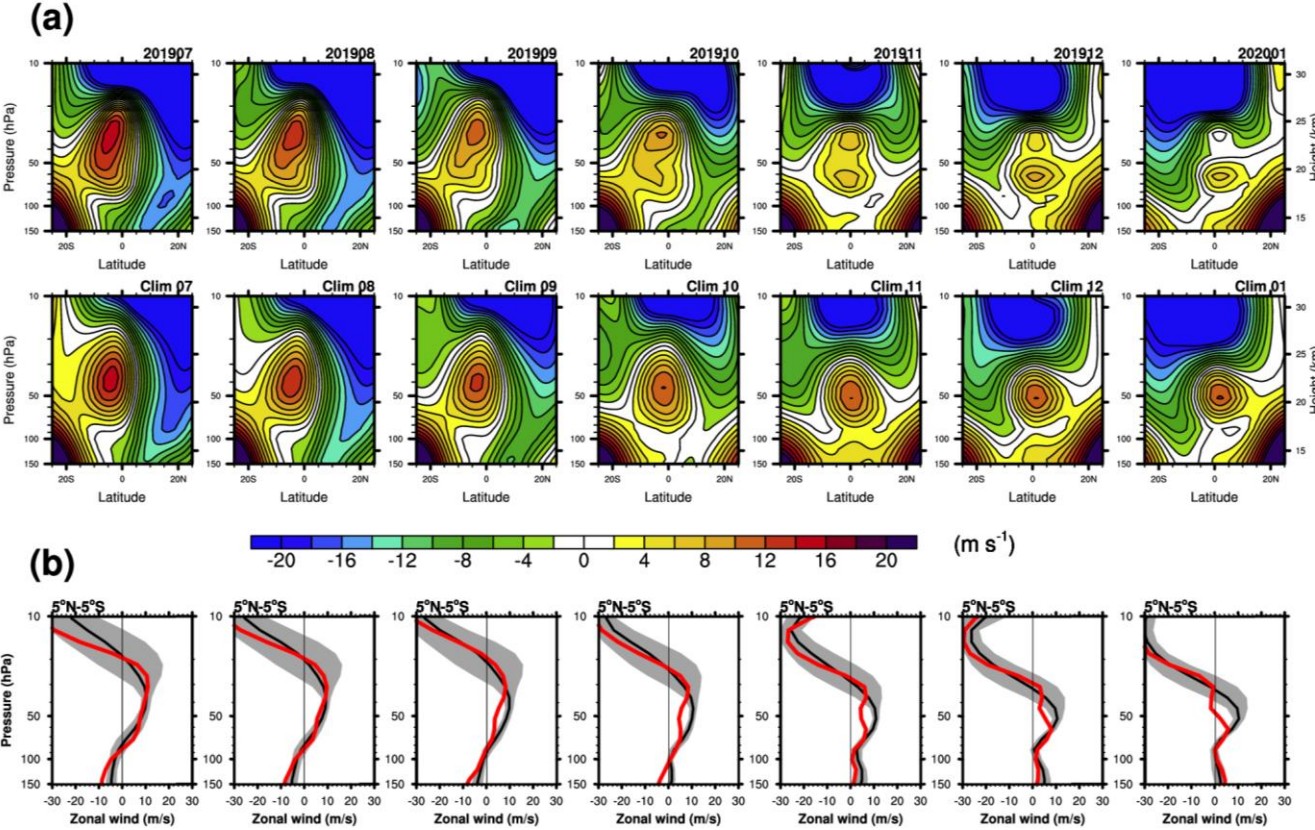

**Figure 1.** (a) (Top) Zonal-mean zonal wind in a latitude–height cross section during July–January for the disruption period (2019/20) and (bottom) the climatology. (b) Zonal-mean zonal wind averaged for 5°N–5°S during July–January for the disruption period (2019/20; red) and the climatology (black) overlaid with the ±1 standard deviation (gray shading). The
climatology corresponds to the westerly QBO years (Sect. 2.1).

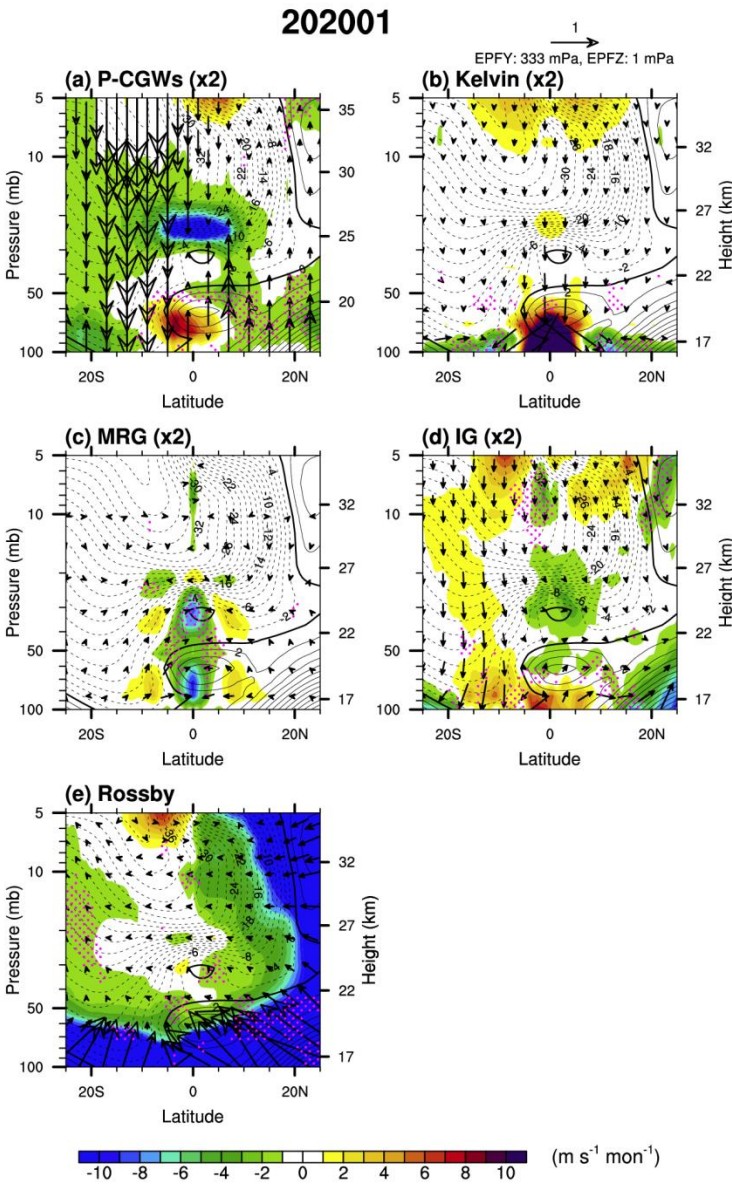

**Figure 2.** EPF (vectors) and EPFD (shading) in a latitude–height cross section for the (a) parameterized CGWs (P-CGWs,
multiplied by 2), (b) Kelvin waves (multiplied by 2), (c) mixed Rossby–gravity waves (MRG, multiplied by 2), (d) inertia–
gravity waves (IG, multiplied by 2), and (e) Rossby waves, overlaid with the zonal-mean zonal wind (contour) in January
2020. Solid (dashed) lines indicate westerly (easterly) winds with an interval of 2 m s⁻¹, and thick solid lines indicate a zero
zonal wind speed. The magenta stipples represent stronger negative EPFD than the climatology by more than its standard
deviation. The reference vector is denoted by an arrow on the upper right corner.




**Figure 3.** Monthly evolution of the (a) zonal-mean zonal wind ($U$), (b) difference in $U$ between the 2019/20 disruption period and the climatology ($U - Uclim$), (c) zonal wind tendency ($\partial U/\partial t$), (d) vertical advection (ADVz), (e) required wave forcing (REQ), and EPFD for the (f) P-CGWs, (g) Kelvin, (h), MRG, (i) IG, and (j) Rossby waves from May 2019 to April 2020 and (k–s) their climatology from May to April from 70 to 10 hPa, superimposed on the zonal-mean zonal wind (black contour lines). The solid (dashed) lines indicate westerly (easterly) winds with an interval of 5 m s$^{-1}$, and thick solid lines indicate a zero zonal wind speed.



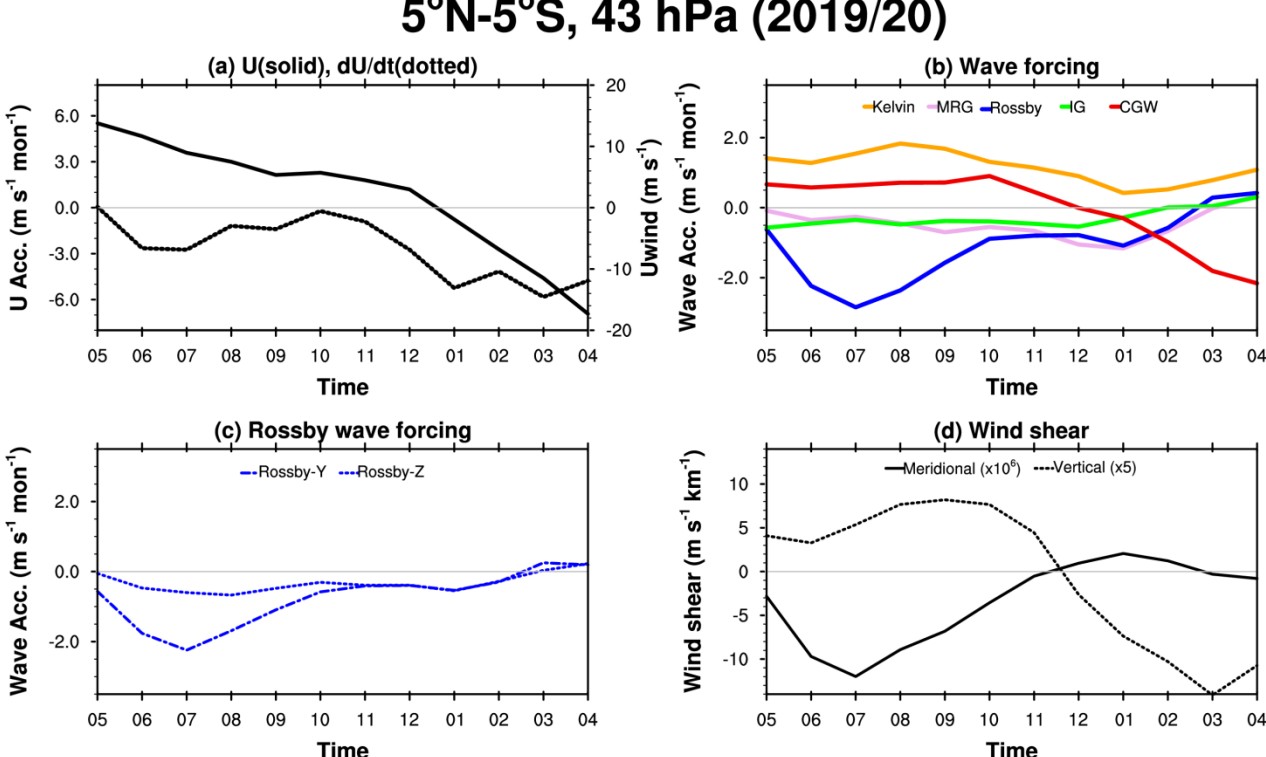

**Figure 4.** Monthly evolution of the (a) zonal-mean zonal wind (solid) and zonal wind tendency (dotted), (b) momentum forcing by the Kelvin waves (orange), MRG waves (pink), Rossby waves (blue), IG waves (light green), and CGWs (red) averaged over 5°N–5°S (dotted) at 43 hPa from May 2019 to April 2020. (c) Momentum forcing by the Rossby waves decomposed into the meridional (dot-dashed) and vertical components (dotted). (d) Meridional wind shear across the equator (solid) and vertical wind shear (dotted).



# Rossby

(m s$^{-1}$ mon$^{-1}$)





**Figure 5.** (first column) EPF divided by air density (vectors) and EPFD (shading) for the Rossby waves in a latitude–height
cross section, along with (second column) their meridional and (third column) vertical components in (a) July 2019, (b)
August 2019, (c) October 2019, and (d) January 2020. The vertical profiles of the meridional EP fluxes at 10°S and 10°N are
presented on the left and right sides of the EPFD-y, and the meridional distribution of the vertical EP flux at 70 hPa is
presented at the bottom of the EPFD-z [red and black lines correspond to the disruption and the climatology, respectively,
with ±1 standard deviation (gray shading)]. The solid (dashed) lines indicate westerly (easterly) winds with an interval of 2
m s⁻¹, and thick solid lines indicate a zero zonal wind speed. The magenta stipples represent stronger negative EPFD than the
climatology by more than its standard deviation.





**Figure 6.** (a) EPF vectors superimposed on the zonal-mean zonal wind (contour) in a latitude–height cross section, (b) vertical component of the EPF at 100 hPa, and (c) zonal-mean zonal wind profile at 15°S in June–July–August (JJA) 2019 (red) and JJA climatology for WQBO (black) with ±1 standard deviation (gray shading).


# MRG



**Figure 7.** EPF divided by air density (vectors) and EPFD (shading) for the MRG waves multiplied by 8 and 4, respectively, in a latitude–height cross section in (a) October 2019, (b) November 2019, (c) December 2019, and (d) January 2020. The vertical profiles of the meridional EP fluxes at 10°S (10°N) are presented on the left and right sides of the EPFD, and the meridional distribution of the vertical EP flux at 70 hPa is presented at the bottom of the EPFD. Contours and the magenta stipples are defined the same as those in Fig. 5.





**Figure 8.** Monthly averaged barotropic instability in a latitude–height cross section (shading) overlaid with the zonal-mean zonal wind (contour) in (a) December 2019 and (b) January 2020. The solid (dashed) lines indicate westerly (easterly) winds with an interval of 2 m s$^{-1}$, and thick solid lines indicate a zero zonal wind speed. Time series of the number of grids with negative daily mean $\overline{q}_\phi$ (s$^{-1}$) in the (c) boxed region in the SH (10°–15°S, 60–90 hPa) in December 2019 and (d) that in the NH (10°–15°N, 60–90 hPa) in January 2020 (red). The black lines in Figs. 8c–d correspond to the climatology with ±1 standard deviation (gray shading).



# Precipitation

(a) 201906

(b) 201907

(c) 201908

(d) 201909

(e) 201910

(f) 201911

(g) 201912

(h) 202001

**Figure 9.** MERRA-2 zonal-mean precipitation in (a–h) June 2019–January 2020 and the climatology (black) overlaid with ±1 standard deviation (gray shading).




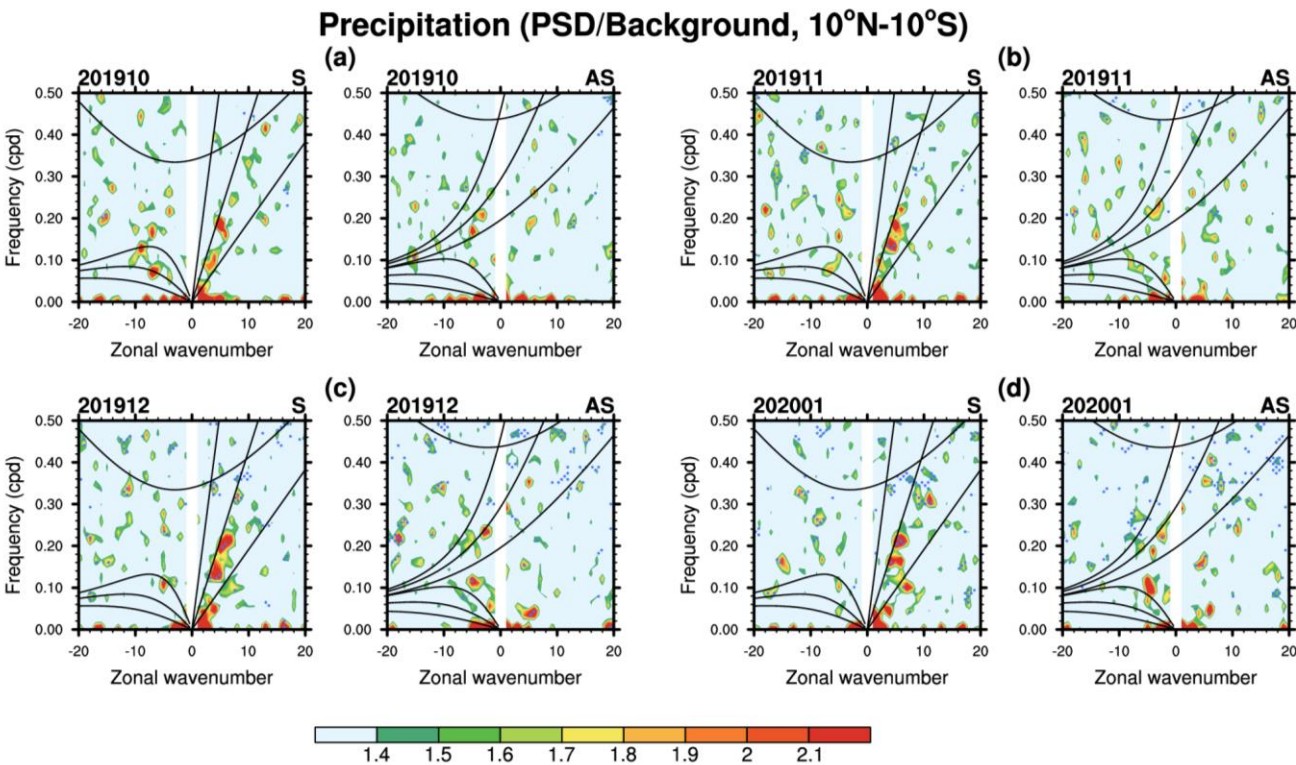

**Figure 10.** Zonal wavenumber–frequency spectra of the MERRA-2 precipitation divided by that of the background spectrum
for (left) symmetric and (right) antisymmetric components, separately, averaged between 10°N and 10°S for (a) October
2019, (b) November 2019, (c) December 2019, and (d) January 2020. The value larger than 1.4 is statistically significant at a
95% confidence level using $t$ test. The blue-stipples represent power-spectral density (PSD) greater than the climatology by
more than its standard deviation. Theoretical dispersion relation for each equatorial wave mode is denoted by black solid line
for the equivalent depth of $\boldsymbol{h}$ = 8, 40, 240 m, but for the IG waves only the $\boldsymbol{h}$ = 8 m line is shown.

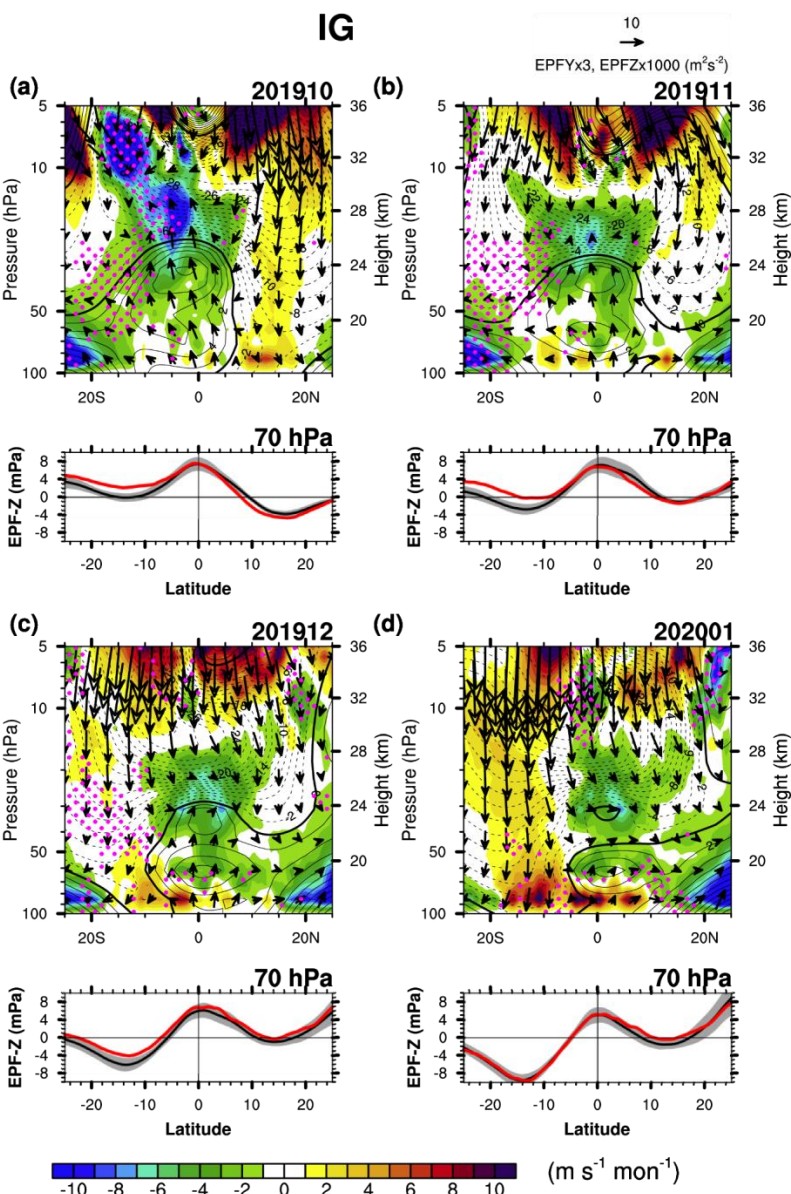

**Figure 11.** EPF divided by air density (vectors) and EPFD (shading) for the IG waves multiplied by 4 in a latitude–height cross section with the (bottom) meridional distribution of the vertical EP flux at 70 hPa in (a) October 2019, (b) November 2019, (c) December 2019, and (d) January 2020 (red) and the corresponding monthly climatology (black) with ±1 standard deviation (gray shading). Contours and the magenta stipples are defined the same as those in Fig. 5.





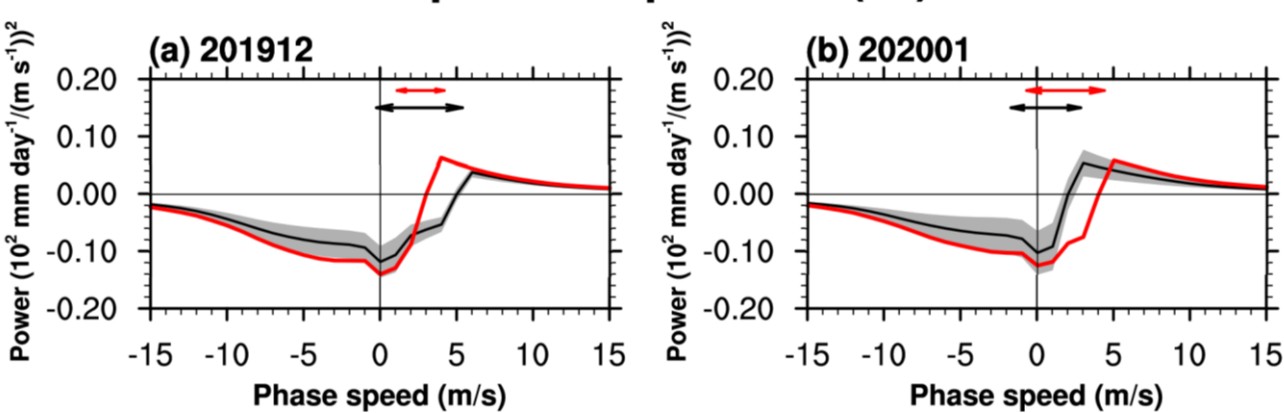

**Figure 12.** 10°S to 10°N averaged precipitation spectrum for the IG wave range [(i) $|k| > 20$ and $\omega > 0$ cpd or (ii) $|k| \leq 20$ and $\omega > 0.4$ cpd] as a function of phase speed in (a) December 2019 and (b) January 2020 along with the corresponding monthly climatology (black) and ±1 standard deviation (gray shading). The spectrum with a negative sign represents the westward-propagating waves. Double-sided arrows in the upper part of each panel indicate the zonal wind ranges between 140 hPa (i.e., source level) and 70 hPa for the QBO disruption period (red) and climatology (black).


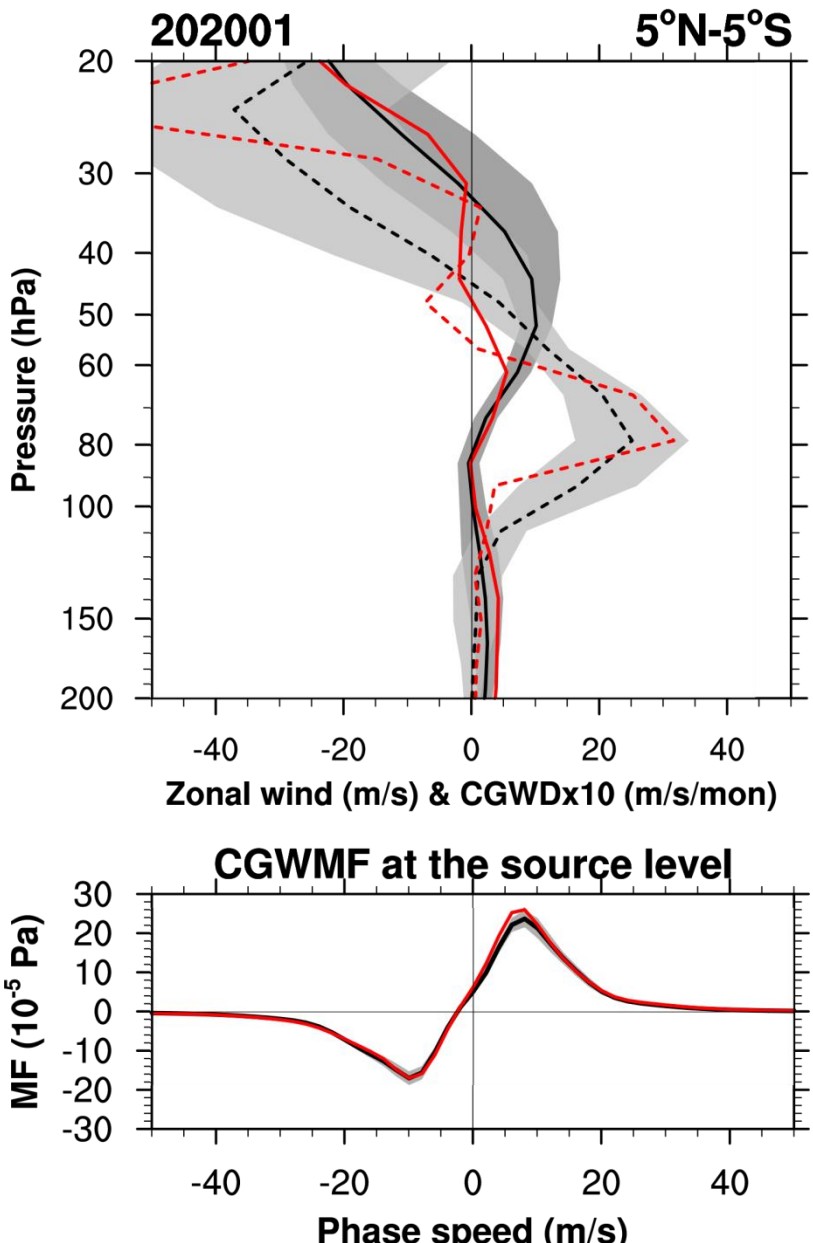

**Figure 13.** (Top) Vertical profiles of the zonal-mean zonal wind (red solid) and zonal-mean CGWD (red dashed) averaged for 5°N–5°S in January 2020 and those for the climatology (black solid and black dashed, respectively) with ± 1-standard deviation (dark-gray and light-gray shading, respectively). (Bottom) Zonal-mean zonal CGW momentum flux spectrum at the cloud top averaged for 5°N–5°S in January 2020 (red) and its climatology (black) with ± 1-standard deviation (gray shading).

**Figure 14.** Phase-speed spectrum of the (left) convective source and (right) wave-filtering and resonance factor (WFRF) in (top) January 2020 and (bottom) and its climatology as a function of the latitude between 20°N and 20°S. Zonal-mean zonal wind at the cloud top ($U_{ct}$) and moving-speed of convection ($c_{qh}$) are denoted by white and gray dashed lines, respectively, in the convective source spectrum.




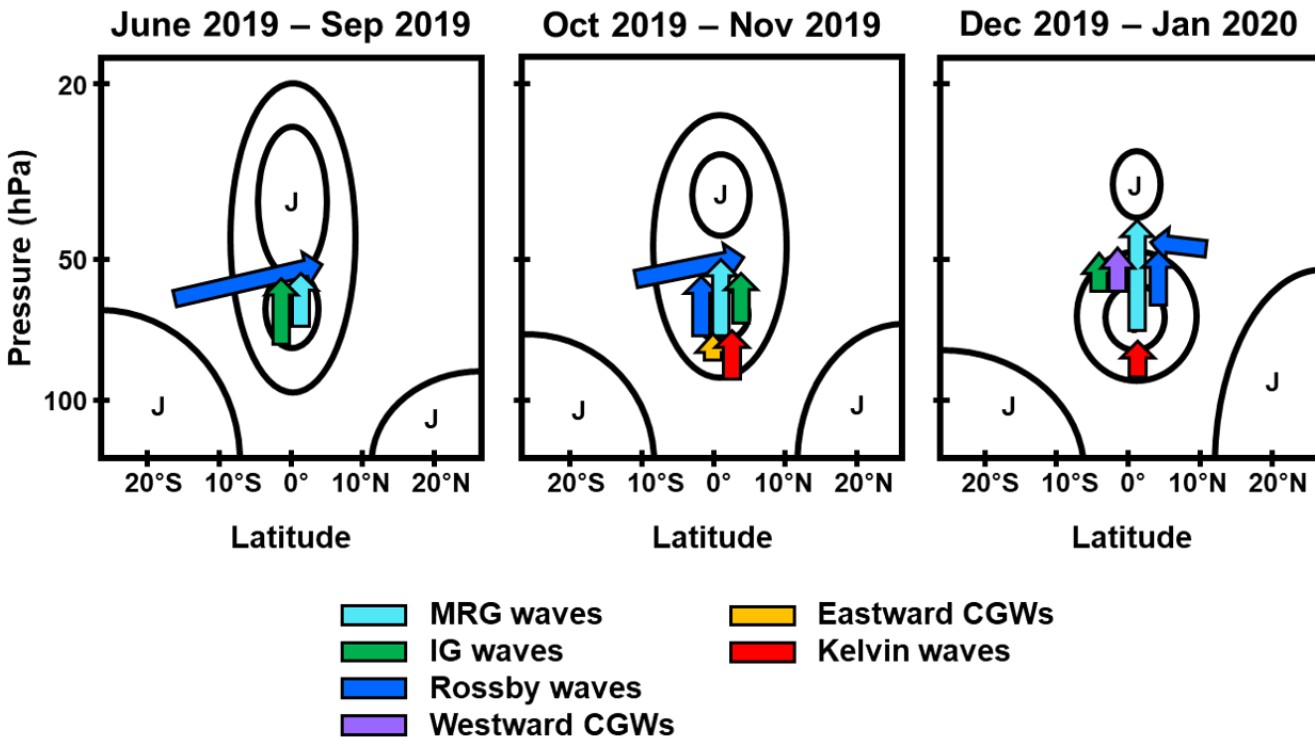

**Figure 15.** Schematic of the zonal-mean zonal wind (black contour) and the wave forcing anomaly compared to the climatology (arrow) during the 2019/20 QBO disruption in June 2019–September 2019 (left), October 2019–November 2019 (middle), and December 2019–January 2020 (right). "J" denotes a westerly jet.