# Peer review of "Contributions of equatorial planetary waves and small-scale convective gravity waves to the 2019/20 QBO disruption Min-Jee Kang and Hye-Yeong Chun"

_Atmospheric Chemistry and Physics, 2021_

## Referee Comment (RC1)

**Review of Kang and Chun (2021) Contributions of equatorial planetary waves and small-scale convective gravity waves to the 2019/20 QBO disruption**

A nice follow-up of the author's 2020 ACP paper about the 2015/16 QBO disruption (KCG20), the submitted manuscript has a very similar structure and the same analysis tools, making a comparison of both events easy to the reader. The manuscript is high quality and I'd even say almost ready for publication.

I only have a couple of general remarks:
- In some places the discussion comparing the 2019/20 to the 2015/16 event can be extended a little, referencing more here and there (see individual comments below).
- Although the authors follow a similar structure to KCG20 and the main differences between both QBO disruptions are summarized in section 4, I think it would be very helpful to add a short subsection at the very end of the results with a discussion about the most striking differences between both QBO disruptions (see the last individual comment for a potential figure). I think this would involve very little restructuring, the figure shouldn't be a lot of work either, and the readers would benefit a lot from having all this information concentrated in one subsection (compared to being spread throughout the whole paper).
–> This should be viewed as just a suggestion, I leave the decision to the authors since the paper is fine anyways in the current format.

**Minor and technical corrections:**

Title: remove 'planetary' since your equatorial wave filters include shorter wavelengths than w1-3.

p.1, l.8: 'that occurred in 2015/16' –> of 2015/16

p.1, l.11: I suggest starting a new paragraph after '… reanalysis data'

p.2, l.55: 'anomalously AND sufficiently strong...'

p.4, l.95-105: very nicely condensed and pointing to KCG20 where needed, good job.

p.4, l.108-109: just to be sure, is this a difference/improvement from KCG20, or still the same?

p.5, l.128: 'shear anomaly and westerly anomaly…' –> do you mean easterly? The red curve is to the left of the climatology July-December at 100-150 hPa

p.5, l.129: 'and January 2020, respectively.' –> remove for clarity + see previous comment.

p.5, l.133-134: 'Close to the equator…' –> it has the lightest shade of green there in Fig 2a, so calling this 'anomalously strong' might be a bit inflated? Or do the authors mean 20-30 hPa?

p.5, l.135-140: I'd be very interested to see this figure for October, where the westerly jet (and the shear zone) at ~30hPa is still strong –> perhaps a good addition to the supplement, to see more detail about the wave forcings in the period leading to the disruption.

Also, the authors note that overall Kelvin wave activity propagating from the troposphere is above average, but still less than the 2015/16 event. Maybe would be worth adding here that there was no strong ENSO this time.

p.6, l.162-163: 'This implies that the ADVz can help QBO disruption…' –> Could the authors elaborate on how this would work? At first, it is a bit counter-intuitive as in principle it should act to increase the WQBO period length, right?

p.6, l.170-172: Towards the end of this paragraph, since the authors compare to the 2015/16 QBO disruption, it would be nice to have some previous references added and perhaps shortly discussed there.

p.6, l.174-180: A short comparison of the dominance (or not) of MRG and IG, to the 2015/16 event would be helpful in this paragraph.
     Also as in the previous comment, a couple of references can be added in the discussion.
     --> A different option: instead of the two previous comments, include an additional paragraph devoted to pointing out the most important differences to 2015/16, among all wave types.

p.6, l.177: '… reversing the sign of the zonal WIND in the later stages.'

p.7, l.202: '… the meridional component becomes weaker,…' ?

Fig. 6c: remove 'latitude' at the bottom, replace with m/s. Also, I recommend that the vertical scale of 6c, and the latitude scale of 6b, are forced to match those of 6a for consistency.

p.9, l.255-260: Maybe add a small comment whether the boxed regions are similar to those of the 2015/16 disruption, since the box in the NH was not included in your previous paper whereas the SH boxes in both studies are close to each other.

(optional) If Fig. 9 doesn't show anything remarkable, I'd move it to the supplement.

p.10, l.298-299: specify / make clearer in this sentence: the IG EPFz 70hPa for the 15/16 case was significantly larger than climatology and the 19/20 case.

p.11, l.356-357: You need to mention ENSO here, as the much more enhanced spectrum in 2015/16 can be partly attributed to the strong el Nino then.

p.12, l.346-347: This is a bit speculative, I think a discussion about lack of el Nino conditions (vs strong El Nino in 2015/16) would help the argument that by elimination global warming may have helped the formation of westerly anomalies, but as you mention later, this needs further research and is beyond the scope of your study.

p.13, l.388-389: One thing that could nicely discern this is doing an additional figure, same as Fig. 4 but for 20-30 hPa, and comparing both 15/16 and 19/20 events.
     I suggest at least adding such a figure into the supplement, but I'd even support making a small subsection in the main manuscript about it - however I leave it up to the authors.

---

## Author Comment (AC1)

**Response to Reviewers' Comments**

**Min-Jee Kang and Hye-Yeong Chun**

**May 6, 2021**

Dear editor and reviewers,

We received two reviews for our manuscript "Contributions of equatorial planetary waves and small-scale convective gravity waves to the 2019/20 QBO disruption" We would like to thank all the reviewers for their helpful and constructive comments. The reviewers' comments made us aware of several important points that should have been addressed. We carefully addressed all comments and tried our best to improve the manuscript based on the suggestions and comments.

During the revision process, we changed the title to "Contributions of equatorial waves and small-scale convective gravity waves to the 2019/20 QBO disruption". In addition, we included (i) a new subsection (Sect 3.7) discussing key differences between 2015/16 and 2019/20 QBO disruptions and (ii) moved Fig. 9 to supplement (Fig. S5), as suggested.

We include a point-by-point response to each comment in the following paragraphs. We indicate the original comment of the respective reviewer in blue color and our answer in black color. In addition, we provide a tracked-changed version of the manuscript.

Sincerely,

Hye-Yeong Chun

**Response to Reviewer #1's Comments**

**General Comment:**

A nice follow-up of the author's 2020 ACP paper about the 2015/16 QBO disruption (KCG20), the submitted manuscript has a very similar structure and the same analysis tools, making a comparison of both events easy to the reader. The manuscript is high quality and I'd even say almost ready for publication.

I only have a couple of general remarks:

- In some places the discussion comparing the 2019/20 to the 2015/16 event can be extended a little, referencing more here and there (see individual comments below).

- Although the authors follow a similar structure to KCG20 and the main differences between both QBO disruptions are summarized in section 4, I think it would be very helpful to add a short subsection at the very end of the results with a discussion about the most striking differences between both QBO disruptions (see the last individual comment for a potential figure). I think this would involve very little restructuring, the figure shouldn't be a lot of work either, and the readers would benefit a lot from having all this information concentrated in one subsection (compared to being spread throughout the whole paper).

-> This should be viewed as just a suggestion, I leave the decision to the authors since the paper is fine anyways in the current format.

Thank you for the reviewer's comments. Following the reviewer's suggestion, a new subsection (Sect 3.7) discussing the key differences between 2015/16 and 2019/20 QBO disruptions is included in the revised manuscript. [p.12, L357–378]

**Minor and technical corrections:**

1) Title: remove 'planetary' since your equatorial wave filters include shorter wavelengths than w1-3.

Thank you for the good suggestion. The title is changed to "Contributions of equatorial waves and small-scale convective gravity waves to the 2019/20 QBO disruption", as suggested. [p.1, L1–2]

2) p.1, l.8: 'that occurred in 2015/16' –> of 2015/16

It is changed as suggested. [p.1, L8]

3) p.1, l.11: I suggest starting a new paragraph after '… reanalysis data'

It is changed as suggested. [p.1, L12]

4) p.2, l.55: 'anomalously AND sufficiently strong...'

It is changed as suggested. [p.2, L55]

5) p.4, l.95-105: very nicely condensed and pointing to KCG20 where needed, good job.

Thank you for your good comment!

6) p.4, l.108-109: just to be sure, is this a difference/improvement from KCG20, or still the same?

This part is the same as in KCG20, which is mentioned in the revised manuscript. [p.4, L114]

7) p.5, l.128: 'shear anomaly and westerly anomaly…' –> do you mean easterly? The red curve is to the left of the climatology July-December at 100-150 hPa

Thank you for your comment. We found that the original sentence was unclear and it is modified in the revised manuscript. [p.5, L132–134]

8) p.5, l.129: 'and January 2020, respectively.' –> remove for clarity + see previous comment.

Thank you. We changed the sentence for clarity. [p.5, L132–134]

9) p.5, l.133-134: 'Close to the equator…' –> it has the lightest shade of green there in Fig 2a, so calling this 'anomalously strong' might be a bit inflated? Or do the authors mean 20-30 hPa?

Thank you for the comment. We intended to say that the negative CGW forcing is stronger than the climatology by more than 1 standard deviation. The sentence is changed in the revised manuscript for clarity. [p.5, L138–139]

10) p.5, l.135-140: I'd be very interested to see this figure for October, where the westerly jet (and the shear zone) at ~30hPa is still strong –> perhaps a good addition to the supplement, to see more detail about the wave forcings in the period leading to the disruption.

Also, the authors note that overall Kelvin wave activity propagating from the troposphere is above average, but still less than the 2015/16 event. Maybe would be worth adding here that there was no strong ENSO this time.

Thank you for your good suggestion! During the revision process, we added a new supplementary figure (Fig. S2) showing all the equatorial wave forcing from July 2019 to February 2020 and found that westerly jet is still strong at ~30 hPa in October along with the evident Kelvin wave forcing at ~40 hPa. In the revised manuscript, we also included a discussion that there was no strong ENSO during the 2019/20 QBO disruption. [Figure S2; p.5, L144–145]

11)      p.6, l.162-163: 'This implies that the ADVz can help QBO disruption…' –> Could the authors elaborate on how this would work? At first, it is a bit counter-intuitive as in principle it should act to increase the WQBO period length, right?

Yes, ADVz generally acts to increase the WQBO period, so the long-lasting WQBO at 20 hPa prevents the westerly QBO to quickly propagate downward. Therefore, the WQBO becomes vertically deep, making the 'middle' WQBO exposed to the negative wave forcing for a long time. More explanation is added in the revised manuscript. [p.6, L169]

12)      p.6, l.170-172: Towards the end of this paragraph, since the authors compare to the 2015/16 QBO disruption, it would be nice to have some previous references added and perhaps shortly discussed there.

Thank you for the comment. The references and short discussion are included in the revised manuscript. [p.6, L178]

13)      p.6, l.174-180: A short comparison of the dominance (or not) of MRG and IG, to the 2015/16 event would be helpful in this paragraph.
Also as in the previous comment, a couple of references can be added in the discussion.
--> A different option: instead of the two previous comments, include an additional paragraph devoted to pointing out the most important differences to 2015/16, among all wave types.

Thank you for your good comment. As suggested last, a new subsection (Sect 3.7) is devoted to discussing the most important differences between 2015/16 and 2019/20 QBO disruptions.

[p.12, L357–377]

14)      p.6, l.177: '… reversing the sign of the zonal WIND in the later stages.

Thank you! It is modified. [p.6, L185]

15)      p.7, l.202: '… the meridional component becomes weaker,…' ?

Thank you for pointing out this error! It is modified. [p.7, L205]

16)      Fig. 6c: remove 'latitude' at the bottom, replace with m/s. Also, I recommend that the vertical scale of 6c, and the latitude scale of 6b, are forced to match those of 6a for consistency.

Thank you for pointing out this part. It is changed as suggested. [Figure 6]

17)      p.9, l.255-260: Maybe add a small comment whether the boxed regions are similar to those of the 2015/16 disruption, since the box in the NH was not included in your previous paper whereas the SH boxes in both studies are close to each other.

In the revised manuscript, we mentioned that a box in the SH is located close to that in KCG20, while one additional box in the NH is considered for the 2019/20 case. [p.9, L263–265]

18)      (optional) If Fig. 9 doesn't show anything remarkable, I'd move it to the supplement.

Thank you for your comment. Fig. 9 of the original manuscript has been moved to supplement (Fig. S5 in the revised manuscript). [Figure S5; p.9, 273–274]

19)      p.10, l.298-299: specify / make clearer in this sentence: the IG EPFz 70hPa for the 15/16 case was significantly larger than climatology and the 19/20 case.

Thank you for the comment. It is modified as suggested. [p.10, L303]

20)      p.11, l.336-337: You need to mention ENSO here, as the much more enhanced spectrum in 2015/16 can be partly attributed to the strong el Nino then.

Thank you. It is included in the revised manuscript. [p.11, L341]

21)      p.12, l.346-347: This is a bit speculative, I think a discussion about lack of el Nino conditions (vs strong El Nino in 2015/16) would help the argument that by elimination

global warming may have helped the formation of westerly anomalies, but as you mention later, this needs further research and is beyond the scope of your study.

I agree with that the original sentence is a bit speculative. Therefore, in the revised manuscript the original sentence is shortened by removing a discussion on El Niño, which is not relevant to the current disruption case. [p.12, L349–351]

22)       p.13, l.388-389: One thing that could nicely discern this is doing an additional figure, same as Fig. 4 but for 20-30 hPa, and comparing both 15/16 and 19/20 events.

I suggest at least adding such a figure into the supplement, but I'd even support making a small subsection in the main manuscript about it - however I leave it up to the authors.

Thank you for the good comment. During the revision process, we plot the same figure as Fig. 4 but for 30 hPa (Fig. A1). It is found that Kelvin wave forcing in December 2019–February 2020 is weaker than that in January–March 2016 (three months centered on each disruption event). However, the magnitude of the Kelvin wave forcing is highly dependent upon the analysis altitude because the location of the positive wind shear changes with time. This implies that it is difficult to compare the magnitude of the positive wave forcing in two disruption cases at a fixed altitude. Although it is rather qualitative, latitude–height structure of the EPD represents better the strength of the positive wave forcing in each month. Therefore, in a new subsection (Sect 3.7), a weaker positive wave forcing during the 2019/20 QBO disruption is explained using latitude–height cross section of the EPD during the 2015/16 QBO disruption (Fig. S3 of KCG20) and during the 2019/20 QBO disruption (Fig. S2 in the revised supplement). However, it is still not clear whether the weak positive wave forcing leads to the vertical wind shear or weak vertical wind shear leads to weak positive wave forcing. The related discussion is also included in the revised manuscript. [p.12, L365–371]

[Figure]

**Figure A1.** Monthly evolution of the momentum forcing by the Kelvin waves (orange), MRG waves (pink), Rossby waves (blue), IG waves (light green), and CGWs (red) averaged over 5°N–5°S (dotted) at 30 hPa from (a) July 2015 to June 2016 and (b) May 2019 to April 2020.

**Response to Reviewer #2's Comments**

**General Comment:**

The paper "Contributions of equatorial planetary waves and small-scale convective gravity waves to the 2019/20 QBO disruption" by Kang et al. investigates which waves contribute to the 2019/20 QBO disruption during the different stages of the disruption. It turns out that in the first phase Rossby waves from the Southern Hemisphere are the leading contribution, and in the later stage tropical MRG waves and Rossby waves from the Northern Hemisphere are the main contributions.

The paper is a follow-up work of a previous paper on the 2015/16 QBO disruption and structured in a similar way for better comparability.

The paper is well written and fits in the scope of ACP and is therefore recommended for publication in ACP after minor revisions.

Thank you! We tried our best to improve the manuscript based on the reviewer's comments.

**Minor Comments:**

1) At the beginning of the introduction you should mention the papers by Ebdon (1960) and Reed et al. (1961), who independently discovered the QBO.

Ebdon, R. A.: Notes on the wind flow at 50 mb in tropical and subtropical regions in January 1957 and in 1958, Q. J. R. Meteorol. Soc., 86, 540-542, 1960.

Reed, R. J., Campbell, W. J., Rasmussen, L. A., and Rogers, R. G.: Evidence of a downward propagating annual wind reversal in the equatorial stratosphere, J. Geophys. Res., 66, 813-818, 1961.

Many thanks for suggesting references. Those are included in the revised manuscript. [p.1, L24–25]

2) Most parameters in Eq.(1) and Eq.(2) are not explained.

Thank you for finding out the mistake. Those are explained in the revised manuscript. [p.3, L87–90]

3) l.95-104: please describe briefly how the k-omega spectra are calculated

It is described in the revised manuscript. [p.4, L101–102]

4) l.129/Fig.1b: The positive wind shear anomaly compared to the climatology at pressures 150-100hPa does no longer hold for January 2020 - in January 2020 the wind shear has a negative anomaly.
And for the other months July-December 2019 at pressures 150-100hPa the wind anomaly is easterly, not westerly, compared to the climatology!

Thank you for your comment. We found that the original sentence was unclear, and it is modified in the revised manuscript. [p.5, L132–134]

5) l.145/146 (Fig.2e): Not clear which Rossby wave forcing you exactly mean. There are no anomalously strong negative values of EPFD at 0-5N in magenta stippled regions. There are several anomalies (magenta stippled regions) which, however, do not really fit to your statement:
5S/50hPa - medium strong EPFD, -2...-4 m/s/month
5N/30hPa - there is only very weak EPFD of -1...-2 m/s/month
10-20N/50-80hPa - very strong EPFD, stronger than -10 m/s/month
Do you mean strong negative, but not anomalously strong EPFD? If yes, why would this be relevant? Please explain in more detail.

Thank you for your comment. The original statement was somewhat unclear. We mentioned Rossby wave forcing at $0°$–$5°$N, 30–40 hPa because the wave forcing at ~40 hPa, which can directly help develop the QBO disruption at 30–50 hPa, is stronger than the climatology (magenta stippled region) even though the magnitude is much smaller than that at 50–80 hPa. The sentence is modified in the revised manuscript. [p.5, L151–153]

6) l.243 / Fig.7: The MRG waves are deduced from antisymmetric k-omega spectra, and in Fig.7 symmetry relative to the equator would be expected. Nevertheless, the EPF and EPFD in Fig.7 are asymmetric. Where does this come from? Because zonal wind and stability are different in the different hemispheres?

Yes, it is because $\bar{u}_y$, $\bar{u}_z$, and $\bar{\theta}_z$ are different in different hemispheres, resulting in asymmetric feature of EPF and EPFD of MRG waves. c.f. In theory, the Kelvin waves have zero meridional perturbation, so the EPF is mainly dominated by $\overline{u'w'}$. This is why the EPF

and EPFD of Kelvin waves show a nearly symmetric feature with respect to the equator.

7) l.265/266: Your statement is not correct!
   The barotropic term is only the second term in Eq.(3), not the first two terms.
   The first term in Eq.(3) is "beta", which is always positive and acts to stabilize the zonal flow. The second term is the barotropic term. If this term dominates and leads to negative dQ/dphi, this is an indication for barotropic instability.

Thank you for your comment. The statement is corrected in the revised manuscript. [p.9, L270–272]

8) l.268-271 / Fig.9: You should mention that the largest difference relative to the climatology is in September 2019! In this month at 5S-5N precipitation is much weaker (by 2...3 sigma!) than the climatology. Do you think this strong anomaly plays a role in the QBO disruption?

I agree with the reviewer's comment. During the revision process, Fig. 9 of the original manuscript is moved to supplementary (new figure as Fig. S5), and the statement is modified in the revised manuscript. [p.9, L273–274]

9) In Fig.10 there are many regions that are above the climatology by more than 1 sigma (blue stippled), but that are at the same time in the light blue range of the color code that is considered insignificant by the t-test.
   Still, these enhancements are discussed as "widening of the spectrum" in December (l.278/279). Can you comment on this?

The reviewer is correct. For the blue stipples to be more meaningful, the ratio between the original spectrum and the background spectrum should be larger than 1.4. In the revised manuscript, the statement is confined to when both conditions are satisfied. [p.10, L282–283]

**Technical Comments:**

1) l.74: The 2019/20 QBO disruption was originally in the westerly QBO phase. -> The QBO was originally in the westerly phase when the 2019/20 QBO disruption happened.

Thank you. It is changed. [p.3, L74]

2) l.76: is greater than -> is more westerly than

It is changed as suggested. [p.3, L76]

3) l.79: downward QBO phase transition -> downward QBO phase progression ??

It is changed as suggested. [p.3, L79]

4) l.92-94: in l.92 Fz consists of three terms! Please check definitions of Fz1 and Fz2! Probably one pair of parentheses is missing in l.92

Thank you for pointing out this error! The definition of the Fz is modified in the revised manuscript. [p.4, L96–97]

5) Fig.4a: the dotted line is hard to distinguish from the solid line

Thank you! The dotted line in Fig. 4a is changed in the revised manuscript. [Figure 4a]

6) Fig.7: for consistency, please add the multiplication factors to the respective panels in Fig.7, similar as in Fig.5d.

Thank you for your comment. Although we respect the reviewer's suggestion, we decided to keep the original version due to the following reasons. First, Fig. 5d is multiplied by a scale factor for a fair comparison with Figs. 5a–c, while all plots in Fig. 7 are multiplied by the same value. Second, since EPFD is multiplied by a factor of 8 and EPF is multiplied by a factor of 4, it is complicated to add the multiplication factors of EPFD and EPF respectively in Fig. 7.

7) caption of Fig.8: barotropic instability -> meridional potential vorticity gradient

It is changed as suggested. [p.27, L584]

8) l.278: The spectrum more than 1sigma stronger than the climatology -> The areas of the spectrum where values are more than 1sigma stronger than the climatology

It is changed as suggested. [p.9, L281]

9) l.280: the strong power is evident in the spectrum related -> areas of strong power that are evident in the spectrum are related

It is changed as suggested. [p.10, L284]

10)    l.314: slightly strong -> slightly stronger

It is changed as suggested. [p.11, L318]

11) l.319/320: The maximum negative CGWD -> At pressures above 40hPa the maximum negative CGWD

It is changed as suggested. [p.11, L323–324]

12) l.369: oval structure of the zonal wind was ... -> the oval-shaped structure of the QBO westerlies that is seen in latitude-altitude cross sections was ...

It is changed as suggested. [p.13, L392]